# Altered Relationship between Functional Connectivity and Fiber-Bundle Structure in High-Functioning Male Adults with Autism Spectrum Disorder

**DOI:** 10.3390/brainsci13071098

**Published:** 2023-07-20

**Authors:** Qiangli Dong, Jialong Li, Yumeng Ju, Chuman Xiao, Kangning Li, Bin Shi, Weihao Zheng, Yan Zhang

**Affiliations:** 1Department of Psychiatry, Lanzhou University Second Hospital, Lanzhou 730000, China; 2Gansu Provincial Key Laboratory of Wearable Computing, School of Information Science and Engineering, Lanzhou University, Lanzhou 730000, China; 3Department of Psychiatry & National Clinical Research Center for Mental Disorders, The Second Xiangya Hospital of Central South University, Changsha 410011, China

**Keywords:** autism spectrum disorder (ASD), magnetic resonance imaging (MRI), structure–function relationships, fiber-bundle microstructures, functional connectivity

## Abstract

Autism spectrum disorder (ASD) is a pervasive neurodevelopmental disorder characterized by abnormalities in structure and function of the brain. However, how ASD affects the relationship between fiber-bundle microstructures and functional connectivity (FC) remains unclear. Here, we analyzed structural and functional images of 26 high-functioning adult males with ASD, alongside 26 age-, gender-, and full-scale IQ-matched typically developing controls (TDCs) from the BNI dataset in the ABIDE database. We utilized fixel-based analysis to extract microstructural information from fiber tracts, which was then used to predict FC using a multilinear model. Our results revealed that the structure–function relationships in both ASD and TDC cohorts were strongly aligned in the primary cortex but decoupled in the high-order cortex, and the ASD patients exhibited reduced structure–function relationships throughout the cortex compared to the TDCs. Furthermore, we observed that the disrupted relationships in ASD were primarily driven by alterations in FC rather than fiber-bundle microstructures. The structure–function relationships in the left superior parietal cortex, right precentral and inferior temporal cortices, and bilateral insula could predict individual differences in clinical symptoms of ASD patients. These findings underscore the significance of altered relationships between fiber-bundle microstructures and FC in the etiology of ASD.

## 1. Introduction

Autism Spectrum Disorder (ASD) is a pervasive developmental disorder characterized by atypical social interaction, communication impairments, and restricted, repetitive, and stereotyped behavior [1]. While ASD is commonly considered a childhood disorder, it can be a lifelong condition that continues to affect individuals into adulthood [2,3]. Prevalence rates of ASD in adults have been estimated to be up to 1–2% [1,4]. However, due to atypical clinical manifestations and psychiatric comorbidity, high-functioning adults with ASD are more prone to misdiagnosis [3,5,6]. In addition, adults with ASD may experience higher rates of depression than the general population, further impacting their mental well-being [1,7]. Given the significant impact on high-functioning adults with ASD, it is crucial to explore more precise diagnostic methods and intervention targets. However, the current understanding of the underlying neural mechanisms in this population is limited.

Existing studies have examined the structural and functional neuroimaging abnormalities in patients with ASD. Regarding structural differences, researchers have observed increased total brain volume in ASD [8], disrupted organization of cortical topology [9,10], as well as enlargements of gray and white matter in the frontal, temporal, and parietal lobes [11,12,13]. Abnormalities in the shape of sylvian fissure, superior temporal sulcus, intraparietal sulcus, and inferior frontal gyrus have also been identified [14,15]. Furthermore, volumes of gray and white matter in individuals with ASD may return to the normal level during development from childhood to adulthood [16,17]. In terms of alterations in brain function, studies have reported a widespread reduction in functional connectivity (FC) in spanning unimodal, transmodal, primary somatosensory, limbic, and paralimbic cortices in patients with ASD compared to the typical developing population [17,18], while increased connectivity has been observed between several subcortical regions [19]. Moreover, the FC in the brain may undergo lifelong changes in the ASD cohort [17].

Diffusion Magnetic Resonance Imaging (dMRI) is a valuable technique for assessing structural connectivity and microstructures in the brain by measuring the diffusion of water molecules in brain tissues and along fiber tracts. Although previous studies have reported changes in structural connectivity and microstructures, such as lower fractional anisotropy (FA) in the left superior thalamic radiation [20], in patients with ASD [21,22], voxel-averaged measures (e.g., FA and mean diffusivity (MD)) do not specifically reflect the changes occurring in fiber tracts. This lack of specificity can confound the interpretation of apparent differences in these metrics [23]. For example, decreased FA could be attributed to various factors, such as abruption along the axons, axonal beading, reduced axonal density, or reduced myelination. Additionally, the diffusion tensor model primarily focuses on the major fiber pathways, ignoring voxels that contain crossing fibers [24,25,26], leading to inaccurate fiber tracking. In contrast, the fixel-based analysis (FBA) approach allows for the investigation of complex microstructures within fiber pathways, even in regions containing crossing-fibers [24,27]. Fixel-based metrics have demonstrated effectiveness as structural biomarkers for ASD diagnosis [28,29]. Therefore, we speculated that FBA metrics can more accurately characterize alterations in fiber connectivity pathways in the ASD population.

Previous studies have primarily investigated structural and functional connectivity as separate entities in the ASD population, without thoroughly exploring their relationship. However, there has been a growing interest regarding the relationship between structural and functional connectivity. Researchers aim to determine if brain function is constrained by structural connectivity and how this relationship changes in individuals with brain disorders [30,31]. For example, previous studies have indicated that the structure–function relationship in patients with ASD was associated with cognitive function, highlighting the significance of functional alignment with structural networks for cognitive flexibility [32,33,34]. Additionally, disruptions in the coupling between structural connectivity and dynamic FC within the rich-club cluster have been observed in patients with juvenile myoclonic epilepsy [35] and classic trigeminal neuralgia [36]. However, these studies relied solely on simple correlational analyses, which may not provide a comprehensive understanding of the complex relationship between structure and function. It is important to note that a substantial portion of the variance in functional connectivity weights cannot be explained by a simple one-to-one correspondence with structural connectivity [37]. Recently, Vázquez-Rodríguez et al., utilized a multivariate linear regression model that incorporated information on spatial proximity, routing, and diffusion between brain regions to predict their functional connectivity [38]. Their findings revealed that the organization of the structural–functional relationship follows a unimodal-to-transmodal gradient, which is closely aligned in unimodal cortex but diverges in transmodal cortex. Consequently, it is necessary to employ an optimized model that takes into account the biological intricacies of the brain to investigate the organization of the structure–function relationship in high-functioning adults with ASD. Nonetheless, previous studies only examined the structure–function relationship from a macroscopic connectivity network perspective, without considering the impact of microstructures within each connectivity pathway on the corresponding FC. Whether the relationship between microstructures and FC follows the similar unimodal-to-transmodal gradient and how it changes in patients with ASD remain unknown.

The present study aims to examine whether ASD is associated with alterations in the coupling between fiber-bundle microstructure and functional connectivity in the brain. We employed a multivariate linear regression model [38] to examine the potential association between FC changes and altered microstructures in the corresponding fiber pathways, as determined through the FBA analysis. Furthermore, we investigated how this relationship between microstructure and function varies across the cortex in high-functioning adults with ASD. We also explored the connection between the altered structure–function relationship and the observed symptoms of this population. By integrating both structural and functional data, our study aims to gain a comprehensive understanding of the structure–function relationship and its relevance to ASD symptoms, thereby offering deeper insights into the neural mechanisms present in high-functioning adults with ASD.

## 2. Materials and Methods

### 2.1. Participants

Participants included in this study were obtained from the Barrow Neurological Institute (BNI) dataset of the ABIDE II database (http://fcon_1000.projects.nitrc.org/indi/abide/abide_II.html, accessed on 7 February 2021). The BNI dataset includes 29 males with ASD and 29 age- and gender-matched typical development controls (TDC), ranging from 18 to 64 years. Participants with an ASD diagnosis were recruited primarily through the Southwest Autism Resource and Research Center, and some participants with ASD and all TDCs were recruited from IRB-approved flyers, word of mouth, media, and local support groups. Written informed consent was obtained through institutional IRB.

Participants with ASD were diagnosed based on the Autism Diagnostic Observation Schedule-2nd edition (ADOS-2) [39], which is a highly recognized evaluative measure for accurately assessing ASD across age, developmental level, and language skills. Participants in the TDC group completed the Social Responsiveness Scale-2nd edition (SRS-2) [40] to screen out significant autism symptoms. The SRS-2 has been frequently used in research settings, which is used to identify the presence and severity of social impairment within the autism spectrum and differentiates it from that which occurs in other disorders. The TDCs also participated in an unstructured detailed interview to ensure the absence of past or current psychiatric or neurological disorders and had no immediate family members with ASD, or other major medical illnesses that would affect brain functioning. Both ASD and TDC participants were right-handed, and had general intellectual abilities measured by the Kaufman Brief Intelligence Test-2 (KBIT-2nd edition) [41] within the range of one standard deviation (SD) below or above the mean. The demographic and clinical information for participants after quality control (see Section 2.3) are given in Table 1.

### 2.2. Imaging Data

Multimodal MRI data (i.e., T1w, rs-fMRI, and dMRI) were downloaded from the ABIDE II database. This dataset was chosen because of its high b-values (2500 s/mm^2^) that allows for more advanced dMRI analysis, e.g., fiber density (FD) [24]. All the images were acquired on a 3.0 Tesla Ingenia scanner. Detailed acquisition parameters are available at the ABIDE II website.

### 2.3. Image Preprocessing

All data were visually inspected for quality control at the time of the scan. The rs-fMRI were preprocessed using the DPARSF toolbox [42] following the default pipeline. Briefly, the first 10 volumes of each image were removed, followed by time-slicing correction, realignment, co-registration to the corresponding T1w images, normalization, detrending, nuisance variables regression (24 head-motion parameters, and white matter, cerebrospinal fluid, and global signals), smoothing using a Gaussian kernel of 6 mm full-width at half-maximum, and bandpass filtering (0.01–0.1 Hz). Participants with head motion over 2.5 mm translation or 2.5 rotation were excluded (3 patients with ASD and 3 TDCs).

The dMRI preprocessing was conducted using MRtrix3 [43]. Images underwent denoising [44], EPI distortion correction, motion and eddy-current corrections, bias field correction [45], and intensity normalization. The FOD image of each subject was computed by using the single-shell three-tissue constrained spherical deconvolution algorithm [46,47]. A population template was generated based on the FOD images, and these images were then registered to the population template using an FOD-guided nonlinear registration algorithm [48]. The population template for each of the 2 groups was also generated for template-level comparison.

### 2.4. Functional and Structural Network Construction

The cerebral cortex was parcellated into 210 regions according to the Brainnetome atlas [49]. A functional network (210 × 210) was constructed by computing the Pearson’s correlation between average time series of pairs of brain regions. Fisher’s r-to-z transformation was applied to improve the normality of FC.

Structural connectivity between brain regions were derived by performing a probabilistic fiber tracking algorithm based on the second-order integration (iFOD2) [50]. This approach addresses the challenge of resolving crossing fibers within a voxel and is more accurate in mapping tractography than a diffusion tensor model. Ten million streamlines were generated across the whole brain based on the default parameters, and the streamlines were filtered to 1 million using spherical-deconvolution-informed filtering of tractograms (SIFT) to reduce tractography bias. A structural network (210 × 210) was generated by counting the number of filtered streamlines between brain regions. The weakest 10% of the connectivity that were considered as spurious streamlines were discarded from the analysis.

### 2.5. Fixel-Based Fiber-Bundle Analysis

Apparent FD and fiber-bundle cross-section (FbC) that reflects the intra-axonal property and fiber-bundle morphology were calculated, respectively [24]. The product of FD and FbC, named FDC, was also used to represent the combined effects of micro- and macro-structures [24]. These features provide more detailed information of fiber tracts within voxels and are more easily interpreted than other voxel-wise features (e.g., fractional anisotropy) [28].

### 2.6. Multilinear Model

A multiple regression model was used to predict the FC profile of each brain region by using microstructures and morphology of fiber tracts connected to this region as predictors (Figure 1). The predictors included average FD and FbC of the shortest path (FD_SP_ and FbC_SP_) between brain regions, and the capacity of information flowing (CIF) between brain regions. Path length was estimated from the weighted structural connectome, which refers to the total sum of individual link lengths that are inversely related to link weights. The CIF between two regions *i* and *j* is defined as follows, with the hypothesis that a fiber tract with higher axonal density, less macroscopic atrophy, and shorter length is more advantageous to information transfer between brain regions:$${\mathrm{CIF}}_{ij}=\frac{1}{N}\sum_{n=0}^{N}{\mathrm{FDC}}_{n}\times \exp(-l_{n}(i,j))$$
where *n* indicates the number of paths between regions *i* and *j*, and *l* is the fiber length of the corresponding path.

The use of multi-variate regression models to relate brain structure and function is similar to [38]. However, the present study focuses on the correspondence between FC and fiber-bundle micro/macro structures rather than structure connectivity. For each region *i*, the following regression model was constructed,
$${\mathrm{FC}}^{i}=\beta_{0}+\beta_{1}{\mathrm{FD}}_{\mathrm{SP}}^{i}+\beta_{2}{\mathrm{FbC}}_{\mathrm{SP}}^{i}+\beta_{3}{\mathrm{CIF}}^{i}$$
where FC*^i^* is a set of functional connections between region *i* and all other regions, and the predictors are FD and FbC of the shortest path, and CIF between region *i* and the other regions in the network. The relationship between fiber-bundle structure and FC of each brain region is quantified by the adjusted R^2^ representing the goodness of fit of the model.

### 2.7. Statistical Analysis

The group comparisons were made at both template and individual difference levels. For template-level comparison, functional networks were averaged across participants of each group to generate a group-level network, whereas structural network and fiber metrics were computed from the population template of each group. The R^2^ values of the whole cortex or within the Yeo’s 7 networks [51] were concatenated to a sequence, and a two sample *t*-test was applied to the sequences to examine the statistical difference between groups. For comparisons at the individual difference level, analysis of covariance (ANCOVA) was performed, with age, intracranial volume, and full-scale IQ as covariates, to examine the between-group difference of R^2^ values at the whole cortex, sub-network, and individual region levels. The reason for choosing these covariates is that they significantly influence the activity and connectivity in the brain [52,53,54], which may bring noises to the analysis (aging effect) and may lead to false positive results that are not truly related to ASD. We also used the ANCOVA model to compare the FC, FD_SP_, FbC_SP_, and CIF between groups. The relationship between regional R^2^ values and clinical assessments (i.e., ADOS and SRS scores) was quantified via the Spearman’s correlation (*ρ*), controlling for the aforementioned covariates. Fisher’s r-to-z transformation was used to determine the significance of between-group differences in correlation coefficients. The false discovery rate (FDR) method at the level of *q* = 0.05 was utilized for multiple comparison correction.

## 3. Results

### 3.1. Demographic and Clinical Characteristics

As shown in Table 1, the average total score of ADOS-2 for the ASD group was 10.92 ± 2.99. There were no significant differences in age and full-scale IQ between the two groups (two sample *t*-test, *p* > 0.1). Compared to the TDCs, patients with ASD showed significantly lower scores in all SRS-2 outcomes (two sample *t*-test, *p* < 0.0001), including the total score and scores of subfields (i.e., awareness, cognition, communication, motivation, and mannerisms).

### 3.2. Comparisons between ASD and TDC at the Template Level

The spatial distribution of R^2^ values is visualized in Figure 2A. The nodal size is inversely proportional to the corresponding R^2^ to highlight regions that have relatively low structure–function correspondence. The distribution pattern of R^2^ values on the cortex is highly organized and spatially symmetric for both ASD and TDC cohorts. Brain regions with high structure–function correspondence mainly include sensorimotor, caudal occipital, medial parietal, and lateral frontal (middle and inferior parts) cortices, whereas the dorsolateral and medial prefrontal cortex, and insular and temporal cortices show less correspondence between FC and fiber-bundle structure. The histogram shows a leftward deviation in R^2^ distribution in patients with ASD relative to the TDCs, with R^2^ values ranging from 0.01 to 0.43 (mean = 0.17) for ASD and from 0.03 to 0.46 (mean = 0.19) for TDC, respectively (Figure 2B).

We then examined the between-group differences of R^2^ values across the whole cortex and within the Yeo’s 7 networks. The structure–function correspondence of the whole cortex was significantly reduced in patients with ASD when compared to the TDCs (*p* = 0.032, Figure 2C). The decreasing trend was also observed in cognitive control network (CCN, *q* < 0.05, FDR corrected) and sensorimotor network (SMN, *p* = 0.01, FDR correction failed) in the ASD cohort (Figure 2D).

### 3.3. Comparisons between ASD and TDC at the Individual Difference Level

The comparison results based on individual differences further supported the findings at the template level. Compared to the TDCs, the mean R^2^ value of the whole cortex in patients with ASD was significantly reduced (*p* = 0.048, Figure 3A), and decreases in R^2^ values in SMN, CCN, and ventral attention network (VAN) were marginally significant (*p* < 0.05, FDR correction failed). Therefore, we performed regional analyses focused only on these three functional networks (i.e., SMN, CCN, and VAN). As shown in Figure 3B, reduced R^2^ values in patients with ASD were found in the left precentral gurus (upper limb region of Brodmann area 4 (A4ul)) (*q* < 0.05, FDR corrected) and right middle frontal gyrus (Brodmann area 46 (A46)) (*p* < 0.0001, FDR correction failed) relative to the TDCs.

We then examined whether the altered nodal structure–function correspondence resulted from changes in FC or the structure of fiber tracts. By comparing the connection profiles of the left A4ul, we found that the FC mainly connecting the parietal and occipital of the right hemisphere (e.g., superior parietal lobule (A5l), paracentral lobule (A1/2/3ll), postcentral gyrus (A2), medial precuneus (A5m), and medial superior occipital gyrus (msOccG)) was significantly decreased in autistic patients (*q* < 0.05, FDR corrected), whereas no significant between-group differences were found in CIF, FD_SP_, and FbC_SP_ between the left A4ul and the aforementioned areas (Figure 3C).

### 3.4. Correlations between Structure–Function Correspondence and Clinical Assessments

Brain regions that showed significantly altered correlations between R^2^ values and SRS scores (Fisher’s z test, *q* < 0.05, FDR corrected) in patients with ASD are visualized in Figure 4A. The alteration of correlations could be categorized into two types: (1) correlations that were insignificant in the TDC group became significant in the ASD group; and (2) the opposite alteration trend. The former included correlations between SRS awareness scores and the R^2^ values of right precentral gyrus (caudal dorsolateral Brodmann area 6 (A6cdl)) (ρ_ASD_ = 0.64, *p*_ASD_ < 0.001; ρ_TDC_ = −0.02, *p*_TDC_ > 0.05) and right inferior temporal gyrus (caudolateral Brodmann area 20 (A20cl)) (ρ_ASD_ = 0.62, *p*_ASD_ < 0.001; ρ_TDC_ = −0.21, *p*_TDC_ > 0.05), respectively, as well as correlations between SRS mannerism scores and the R^2^ values of right inferior temporal gyrus (extreme lateroventral Brodmann area 37 (A37elv)) (ρ_ASD_ = 0.66, *p*_ASD_ < 0.001; ρ_TDC_ = −0.01, *p*_TDC_ > 0.05). The latter included correlations between R2 values of left superior parietal lobule (intraparietal Brodmann area 7 (A7ip) and rostral Brodmann area 7 (A7r)) and SRS cognition (ρ_ASD_ = −0.10, *p*_ASD_ > 0.05; ρ_TDC_ = 0.61, p_TDC_ < 0.001) and mannerism scores (ρ_ASD_ = −0.19, *p*_ASD_ > 0.05; ρ_TDC_ = 0.66, *p*_TDC_ < 0.001), respectively.

We also found a significant association between structure–function correspondence of insular cortex and ADOS scores in the ASD group. Specifically, R^2^ values of right dorsal dysgranular insula (dId) were positively correlated with ADOS social and stereotyped behavior scores (ρ = 0.61 and 0.69, respectively, and *p* < 0.001); and R^2^ values of left hypergranular insula (Ih) were negatively correlated with ADOS creativity scores (ρ = −0.74, *p* < 0.0001).

## 4. Discussion

The aim of the present study was to investigate whether individuals with ASD exhibit changes in the macroscale hierarchies of relationships between fiber-bundle structure and FC throughout the cortex. Our results suggested that high-functioning adults with ASD were characterized by (1) an untethered structure–function relationship in both unimodal and transmodal cortices (e.g., SMN and CCN); and (2) these changes were able to predict individual differences in clinical symptoms among the patients. To the best of our knowledge, this study provided the initial evidence for altered connections between microscopic fiber-bundle structures and functional connectomes in patients with ASD, which might to some extent promote our understanding of the structural and functional basis underlying impaired social cognition and behavior in this population.

In this study, we employed a multiple regression model inspired by a previous study [38] to examine the structure–function relationship of each brain region. However, we took a step further by representing connections in structural networks using the fixel-based metrics (i.e., FD, FbC, and FDC), which have been demonstrated as effective markers for characterizing structural abnormalities of fiber tracts in autistic populations [28,29]. Moreover, weighting the edges through these metrics could also preserve the topological organization of a network. Our findings suggest that the relationship between functional connectomes and fiber-bundle structures may not be uniform across the cerebral cortex in both individuals with ASD and the TDCs. By using the fixel-based metrics, we were able to capture these subtle differences and gain further insights into the complex relationship between structure and function in the context of ASD.

### 4.1. Divergent Structure–Function Relationships in Patients with ASD

The structure–function relationships of both ASD and TDC cohorts were strongly aligned in the unimodal cortex but untethered in the transmodal cortex. This phenomenon is in line with previous findings on the healthy population [38], which may be related to the established structure network architecture, e.g., brain regions demonstrating strong structure–function coupling were found to be primarily associated with local white matter pathways, while regions with weaker coupling rely more on indirect long-range structural connections [38,55]. However, compared to the TDCs, individuals with ASD showed a significant reduction in structure–function coupling throughout the entire cortex, both at the template and individual levels. The decreased coupling may suggest a disorder in structural and/or functional organization of the cortex in patients with ASD. Interestingly, while changes in functional connectivity were observed in the ASD group, no significant differences were found between the two groups in terms of fixel-based metrics. This suggested that the altered structure–function correspondence in patients with ASD may primarily be attributed to abnormalities in the functional connectome, rather than the underlying white matter connectivity. These findings highlighted the importance of investigating both the functional and structural aspects of brain connectivity to comprehend the complex nature of altered brain networks in ASD.

In addition, we observed a significant decrease in the structure–function relationship originating from the A4ul cortex in patients with ASD compared to the TDCs, and the abnormal structure–functional correspondence observed in the A4ul structure primarily stems from alterations in FC. Furthermore, the FC abnormality of the A4ul cortex of individuals with ASD involves connections with the primary sensory cortex, somatosensory association cortex, and visual cortex. The A4ul is known to be associated with sensorimotor behavioral processes, vision, and dementia [56,57,58]. Previous studies have reported reduced long-range FC in ASD, including decreased connectivity between the primary motor and visual cortex [59,60,61,62]. Our findings were consistent with these previous studies. The decreased connectivity between the primary motor and visual cortex, as observed in our study and reported in previous research, could potentially contribute to a difficulties in integrating visual information with motor processes in the ASD cohort [59,60,61,62]. Interestingly, while changes in functional connectivity were observed in the ASD group, no significant differences were found between the two groups in terms of fixel-based metrics. This suggested that the altered structure–function correspondence in patients with ASD may primarily be attributed to abnormalities in the functional connectome, rather than the underlying white matter connectivity. These findings highlighted the importance of investigating both the functional and structural aspects of brain connectivity to comprehend the complex nature of altered brain networks in ASD.

### 4.2. Autistic Clinical Symptoms Were Related to Subtle Changes in Structure–Function Relationships

In the present study, the ASD group exhibited a significant correlation between the R^2^ values and the SRS scores in the A6cdl and inferior temporal gyrus. Additionally, a significant correlation was found between the R^2^ values and ADOS scores in the insular cortex. These findings suggested potential differences in the coupling between structure and function of these regions between the two groups. These findings were consistent with previous research indicating that the structure–function coupling of the brain regions in the unimodal, paralimbic, and idiotypic primary cortex were related to the clinical symptoms of ASD [63]. Several prior studies have reported abnormal FC in the A6cdl and inferior temporal gyrus in individuals with ASD [64,65,66]. Therefore, a possible explanation for our results could be that, when an external input is received, the FC in these brain regions may significantly increase or decrease, resulting in a discrepancy between functional connectivity and underlying structural connectivity. Consequently, this disruption in the correspondence between brain structure and function may contribute to the manifestation of symptoms related to social responsiveness and autistic behaviors.

On the other hand, in the left superior parietal lobule, no noteworthy correlation was discovered between the R^2^ values and the SRS scores in the ASD group. Conversely, the TDC group exhibited an opposite inclination, suggesting that the relationship between structure–function coupling and clinical symptoms might vary among different brain regions between the two groups. One plausible explanation for this discrepancy is that the heterogeneity of ASD individuals [67] may lead to variations in brain structures, functions, and symptom manifestations within the ASD group. In contrast, the TDC group may exhibit a more consistent relationship between brain structure, function, and symptomatology.

### 4.3. Limitations

There were several limitations to the current study. First, the sample size of this study was limited. The FBA analysis required a high b-value, over 2000 s/mm^2^, which led to the exclusion of almost all the data in the ABIDE database except for the BNI dataset. This is a strength in the sense that the FBA analysis could capture more detailed microstructural changes. However, the small sample, especially with such a broad age range (18–64 years), may influence the analysis results, though we have controlled for the effect of age. Therefore, it is still necessary to validate our findings on larger independent datasets in order to further examine their validity, reproducibility, and generalizability. Second, although the gradient of the structure–function relationship we showed on the cerebral cortex was in a good accordance with previous literature [38], these results were derived from different diffusion metrics. The influence of using different anatomical features on the structure–function relationship will be explored in future work. Third, the BNI dataset only includes male participants. Gender differences have been shown to influence neurodevelopmental trajectories of the brain regarding brain volume and cerebral blood flow, etc., and ASD-related brain changes may vary between males and females [68]. Although only using male participants could fully eliminate the gender effects on statistical analysis, including female participants may help to detect the homogeneity and heterogeneity of alterations in the brain between sexes. This factor will be considered in future work.

## 5. Conclusions

In conclusion, our study revealed that the relationships between fiber-bundle microstructures and FC followed a similar spatial gradient in both ASD and TDC, i.e., corresponded closely in the unimodal cortex but diverged in the transmodal cortex. We also found an increased divergence between structure and function in patients with ASD. Additionally, the structure–function relationships of certain brain regions could predict individual differences in clinical symptoms, despite the absence of significant between-group difference. These findings provided direct evidence for the abnormal structure–function relationships in the ASD cohort, which could potentially facilitate the clinical assessment of ASD.

## Figures and Tables

**Figure 1 brainsci-13-01098-f001:**
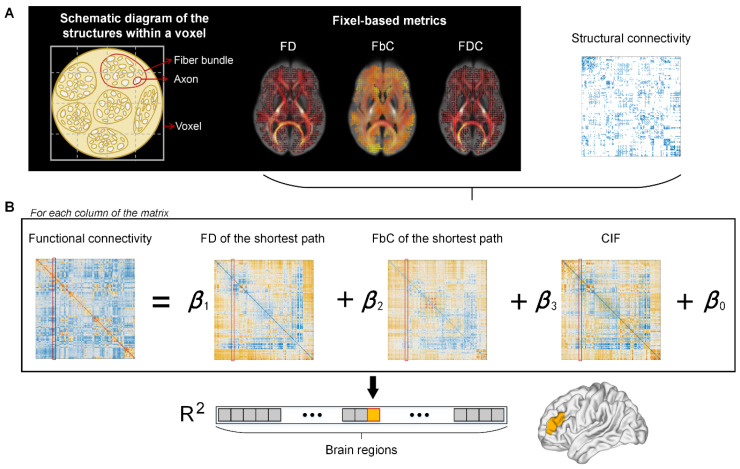
The pipeline for calculating node-wise structure–function relationship. (**A**) The fixel-based metrics, including fiber density (FD), fiber-bundle cross section (FbC), and the combination of these two measures (FDC), are used as weights for edges in the structural network. (**B**) The structure–function relationships were estimated by fitting a multilinear regression model for each node, separately. The predictor and independent variables are the structural and functional relationships between the target node and all the other nodes, respectively, including the shortest path length of FD and FbC networks, the capacity of information flowing (CIF), and the functional connectivity. Goodness of fit for each node is quantified by R^2^ value between observed and predicted functional connectivity.

**Figure 2 brainsci-13-01098-f002:**
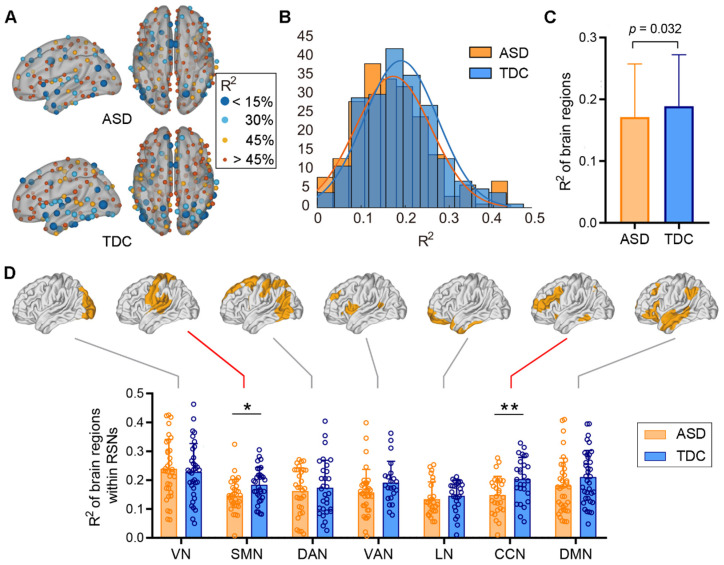
Comparison of the structure–function relationships across the cortex between ASD and TDC at the template level. We generated a population template for each group to compare the between-group difference at the template level. (**A**) The spatial distribution of the structure–function correspondence of the two groups. Nodes are colored and sized in inverse proportion to R^2^. (**B**) The histograms of the R^2^ values across brain regions in the two groups. (**C**) The R^2^ values of the entire cortex significantly decrease in ASD group relative to the TDCs (two sample *t*-test, *p* = 0.032). (**D**) Significant reductions in the R^2^ values in cognitive control and sensorimotor networks in the ASD cohort relative to the TDCs (two sample *t*-test, ** *q* < 0.05, FDR corrected, * *p* = 0.01, uncorrected).

**Figure 3 brainsci-13-01098-f003:**
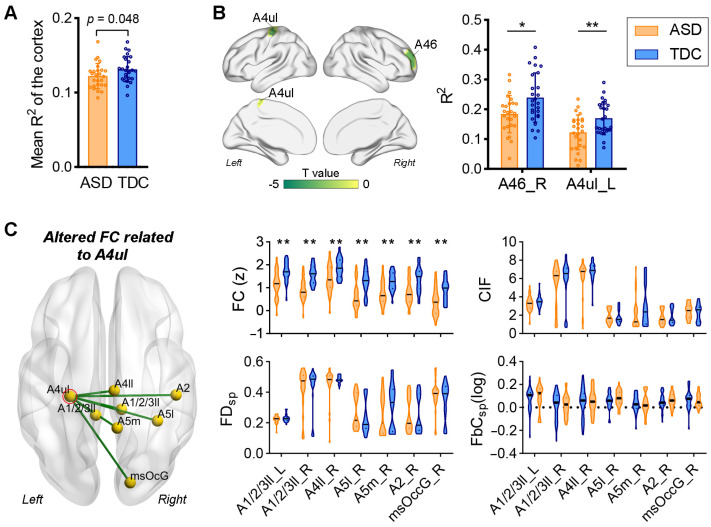
Comparison of the structure–function relationships across the cortex between ASD and TDC at the individual level. We directly compared the R^2^ values between individuals of the two groups. (**A**) The R^2^ values of the entire cortex significantly decrease in the ASD group relative to the TDCs (ANCOVA, *p* = 0.048). (**B**) Comparison of regional R^2^ values between ASD and TDC. Significant decreases in R^2^ values in the A4ul (ANCOVA, ** *q* < 0.05, FDR corrected) and A46 (ANCOVA, * *p* < 0.0001, FDR correction failed). (**C**) Comparisons of structural and functional connections related to the A4ul between ASD and TDC. Significantly reduced functional connectivity (FC) between the A4ul and seven brain regions can be observed in the ASD cohort (ANCOVA, ** *q* < 0.05, FDR corrected), whereas no significant between-group differences were found in structural measures of these connections.

**Figure 4 brainsci-13-01098-f004:**
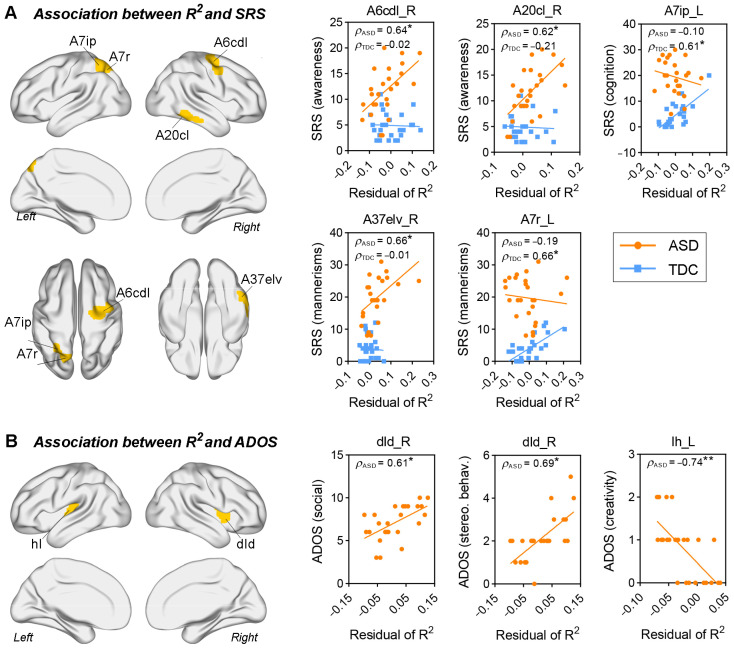
Correlation between the regional R^2^ values and clinical assessment. (**A**) Brain regions with significant between-group differences in the correlations (Fisher’s z test, * *q* < 0.05, FDR corrected). (**B**) Brain regions that have the R^2^ values significantly correlated with ADOS scores (* *p* < 0.001, ** *p* < 0.0001). Age, intracranial volume, and full-scale IQ were regressed out from the original R^2^ values by using a linear regression model. The residuals of R^2^ were used for correlation analysis.

**Table 1 brainsci-13-01098-t001:** Demographic and clinical information of participants.

	ASD	TDC	*p* Value
Age	38.15 ± 16.18	40 ± 15.35	0.67 ^a^
Gender	26 male	26 male	-
Full-scale IQ	109.31 ± 13.66	111.27 ± 12.18	0.58 ^a^
ADOS			
Communication	3.80 ± 1.32	-	-
Social	7.12 ± 2.01	-	-
Stereotyped behavior	2.12 ± 1.09	-	-
Creativity	0.80 ± 0.71	-	-
Total	10.92 ± 2.99	-	-
SRS (raw scores)			
Awareness	11.68 ± 4.71	4.92 ± 2.46	<0.0001 ^a^
Cognition	19.52 ± 6.60	4.54 ± 4.62	<0.0001 ^a^
Communication	35.64 ± 11.51	6.65 ± 5.82	<0.0001 ^a^
Motivation	19.16 ± 6.39	5.77 ± 4.49	<0.0001 ^a^
Mannerisms	19.76 ± 6.63	4.00 ± 3.64	<0.0001 ^a^
Total	105.76 ± 31.92	25.88 ± 16.71	<0.0001 ^a^

Abbreviations: ASD, autism spectrum disorder; TDC, typical developmental control; IQ, intelligence quotient; ADOS, Autism Diagnostic Observation Schedule; SRS, Social Responsiveness Scale. Note: Values are mean ± SD. Two ASD subjects, one of whom lacked scores of module 4 of ADOS-2 and another who lacked SRS assessment, were excluded from the clinical phenotype-related analysis. ^a^ Two sample *t*-test.

## Data Availability

We thank the numerous contributors to the ABIDE database for their effort in the collection, organization, and sharing of their datasets. The data that support the findings of this study are openly available in the ABIDE at http://fcon_1000.projects.nitrc.org/indi/abide/abide_II.html (accessed on 7 February 2021).

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
