# Peer review of "Altered Relationship between Functional Connectivity and Fiber-Bundle Structure in High-Functioning Male Adults with Autism Spectrum Disorder"

_brainsci, 2023, doi:10.3390/brainsci13071098_

Round 1
Reviewer 1 Report
The authors describe a study aimed at investigating the relationship between functional connectivity and fiber-bundle structure in high-functioning male adults with autism spectrum disorder. The topic is of interest since the underpinning neurophysiological mechanisms of ASD condition is unclear, even though numerous studies on functional or structural connectome alterations are reported in literature.
Since many studies reported gender differences in ASD condition, given that the study was conducted on male individuals only, I recommend to slightly modify the title by specifying “high functioning male adults”
The described preprocessing steps for the extraction of functional and structural connectivity are in line with the literature and are well reported. However, I believe that some additional details should be added to the statistical methods. In particular, it should made clear that for each individual and for each brain region a multilinear model was fitted. Moreover, since one of the most important findings is the reported correlation with some clinical test scores, the authors should clarify what they mean when they refer to the “residuals of R2” (figure 4), a variable not described in the Methods section.
Regarding the interpretation of the results, I’m not sure that the study supports the sentence reported at lines 347-349 “we proposed that the observed structure-function uncoupling in both unimodal and transmodal cortex in individuals with ASD could be attributed to the abnormal development of local white matter”. The fact that there is a weaker coupling in ASD subjects between structural and functional connectivity in some areas, and considering at the same time that alterations at group level could be identified on FC only (Figure 3), does not seem in my opinion to indicate an “abnormal development of local white matter” as the cause.
As properly reported in the Limitation section, the sample size is very small. The authors should temper the discussion on the findings by using less conclusive terms.
Minor points
I suggest to modify the colors in the brain maps in figure 2D, 3B, 3C in order to facilitate the reader that in all the manuscript is used to interpret the orange and blue colors as referred to ASDs and TDCs, respectively.
Typo at line 289: later-latter
Typo at line 296: ASOS-ADOS
Line 397: I suggest to use the term “Therefore” instead of “However”
Reviewer 2 Report
This is an interesting study examining the relationship between functional connectivity and fiber-bundle structure in high-functioning adults in autism spectrum disorder. The paper is well-written and easy to follow. I agree that it may contribute to the literature well. I only have a few comments to improve the manuscript further:
1. I appreciate the fact that the inclusion and exclusion criteria of the study is well-described. This is commendable. I hope the authors can also provide more information on how the participants were recruited?
2. It was mentioned that participants with ASD were diagnosed based on the Autism Diagnostic Observation Schedule-2nd edition (ADOS-2) and that participants in the TDC group completed the Social Responsiveness Scale-2nd edition (SRS-2). It would be helpful to provide a brief explanation of these diagnostic and assessment measures to ensure readers understand their significance and reliability.
3. It was mentioned that the participants' age range is from 18 to 64 years. It might be worth discussing how the age range was determined and whether there were any specific considerations regarding the inclusion/exclusion criteria related to age. Also, the age range seems to be huge, which may raise some concern.
4. It might be worthwhile to discuss how the use of only male participants may affect the results and its generalizability.
5. Participants' demographic and clinical information is given in Table 1 after quality control. It would be beneficial to describe the specific criteria and procedures used for quality control to ensure the reliability and validity of the data.
6. It would be beneficial to mention whether the FDR correction was applied to the t-tests, ANCOVA, or both.
7. It was mentioned that age, intracranial volume, and full-scale IQ were used as covariates in the ANCOVA analyses. It would be useful to briefly discuss the rationale for selecting these specific covariates and how they might influence the interpretation of the results.
8. It might be helpful to elaborate further the clinical implication of the current findings.
Reviewer 3 Report
This is a very interesting study investigating the relationship between functional connectivity and structure in high-functioning adults suffering from autism spectrum disorder. The paper is well-written and of interest for the journal; however, several minor changes should be made before considering it for publication.
ABSTRACT
1- I recommend to clarify what does "how they are coupled" means. This terms can be confusing for the readers.
2- The main study design is not adequately described in the abstract section. How were patients recruited?
3- The main aims of the paper should be described in the first lines of the abstract.
INTRODUCTION
1- At the end of the introduction the authors report "In the present study, we employed a multi-variate linear regression model". The authors are describing the study design and not the main aims. I recommend to expand this section by describing the main goals. A separate subsection would be recommended (1.1. Aims).
MATERIALS AND METHODS
1- I recommend to add a section called "Participants and study design". The authors start this section by describing Imaging data, image preprocessing.
2- Demographic and clinical characteristics of the sample should be better described in the Results section.
RESULTS
1- What does "Template level" mean? Does it mean "baseline"?
2- The title of the subsection 3.3. is too long and confusing.
DISCUSSION
1- A conclusions section is needed.
2- The authors report limitations. They should also report for strenghts.
Round 2
Reviewer 2 Report
The authors have addressed all my comments well. I appreciate all their efforts.